# ITERATIVE TASK-ADAPTIVE PRETRAINING FOR UNSUPERVISED WORD ALIGNMENT

## ABSTRACT

How to establish a closer relationship between pre-training and downstream task is a valuable question. We argue that task-adaptive pretraining should not be just performed before task. For word alignment task, we propose an iterative self-supervised task-adaptive pretraining paradigm, tying together word alignment and self-supervised pretraining by code-switching data augmentation. When we get the aligned pairs predicted by the multilingual contextualized word embeddings, we employ these pairs and origin parallel sentences to synthesize code-switched sentences. Then multilingual models will be continuously finetuned on the augmented code-switched dataset. Finally, finetuned models will be used to produce new aligned pairs. This process will be executed iteratively. Our paradigm is suitable for almost all unsupervised word alignment methods based on multilingual pre-trained LMs and doesn't need gold labeled data, extra parallel data or any other external resources. Experimental results on six language pairs demonstrate that our paradigm can consistently improve baseline method. Compared to resource-rich languages, the improvements on relatively low-resource or different morphological languages are more significant. For example, the AER scores of three different alignment methods based on XLM-R are reduced by about $4 \sim 5$ percentage points on language pair En-Hi.

## 1 INTRODUCTION

Although pre-trained language models (PTLMs) (Devlin et al., 2019b; Conneau et al., 2020)trained with massive textual and computational resources have achieved high performance in natural language processing tasks, there can be a distributional mismatch between the pretraining and target domain corpora. To tackle domain discrepancies, domain-adaptive pretraining with a large corpus in the domain of the downstream task is usefully employed, such as BioBERT (Lee et al., 2020). However, this approach requires large corpora in the target domain and entails a high computational cost. Gururangan et al. (2020) propose task-adaptive pretraining and explore the benefits of continued pretraining on data from the task distribution. There are also others works (Gu et al., 2020; Karouzos et al., 2021; Nishida et al., 2021) focusing on establishing a closer relationship between pre-training and downstream task. For example, Gu et al. (2020) add a task-guided pre-training stage with selective masking between general pre-training and fine-tuning. Karouzos et al. (2021) simultaneously minimize a task-specific loss on the source data and a language modeling loss on the target data during fine-tuning.

However, these methods generally follow a fixed paradigm: task-adaptive pretraining then task training. There is an obvious lack of interactive feedback. Can the output of the task can be used to improve pretraining? See Figure 1. And we find that an iterative self-supervised task-adaptive pretraining paradigm can be designed for unsupervised word alignment tasks. In the following, we give a detailed introduction about how to design our new paradigm.

Continued pretraining of a LM on the unlabeled data of a given task (task-adaptive pretraining) (Gururangan et al., 2020) has been shown to be beneficial for task performance. And we think that simply pre-training LMs with MLM or TLM on monolingual parallel sentences is obviously not closely integrated with word alignment task. Based on the assumption that a closer interaction between task pre-training and the task itself can improve performance, we propose an iterative self-supervised continued pretraining paradigm, constantly pushing pre-trained LMs toward the word

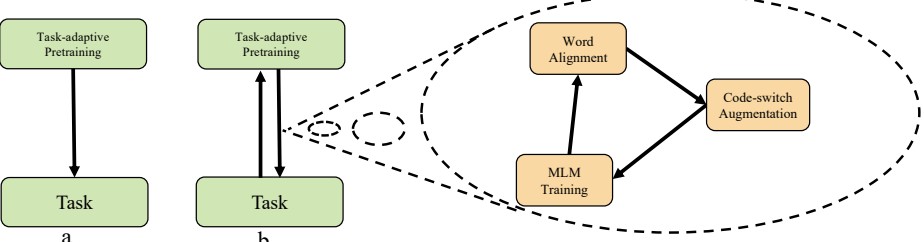

Figure 1: Continued pretraining of a LM on the unlabeled data of a given task (task-adaptive pretraining) (Gururangan et al., 2020) has been show to be beneficial for task performance. However, it only occurs before tasks and is obviously not closely integrated with tasks. For word alignment task, we propose a new paradigm, in which task-adaptive pretraining and word alignment be executed iteratively.

alignment task. We augment sentences with self-labeled pairs and code-switching strategy. When we get the aligned pairs predicted by the multilingual contextualized word embeddings, we employ these pairs and origin parallel sentences to synthesize code-switched sentences. Then multilingual models will be continuously finetuned on the augmented code-switched dataset. Finally, finetuned models will be used to produce new aligned pairs. This process will be executed iteratively. On the one hand, data augmentation with self-labeled pairs and code-switching strategy can alleviates data scarcity. On the other hand, training LMs with MLM on code-switched sentences can bring the different languages closer together in the embedding space and if we training LMs on both code-switched sentence and corresponding origin sentences, the code-switched tokens (predicted pairs) will also move towards each other in the embedding space.

Our contribution can be listed as follows:

- We design an iterative task-adaptive pretraining paradigm for word alignment, in which task-adaptive pretraining will be performed not only before task but also after task. In other words, task-adaptive pretraining and word alignment will be executed iteratively.

- Our paradigm is suitable for almost all unsupervised word alignment methods based on multilingual pre-trained LMs and doesn't need gold labeled data, extra parallel data or any other external resources. We also don't need to introduce carefully designed loss function and the results can be easily reproduced.

- In-depth analysis reveals that training model with standard masked language modeling on source-language,target-language and code-switched sentences is approximately optimizing the similarity of switched tokens. This can serve as a potential explanation why code-switching can be used to improve machine translation and cross-lingual tasks.

- We perform experiments on six language pairs and demonstrate that our paradigm can consistently improve baseline methods. For example, the AER scores of three different alignment methods based on XLM-R are reduced by about $4 \sim 5$ percentage points on language pair En-Hi.

## 2 RELATED WORK

### 2.1 TASK ADAPTIVE PRETRAINING

Language models pretrained on text from a wide variety of sources form the foundation of today's NLP. Gururangan et al. (2020) propose domain-adaptive pretraining and task-adaptive pretraining. They explore the benefits of continued pretraining on data from the task distribution and the domain distribution. Gu et al. (2020) add a task-guided pre-training stage with selective masking between general pre-training and fine-tuning. Jin et al. (2022) study a lifelong language model pretraining challenge where a PTLM is continually updated so as to adapt to emerging data. Karouzos et al. (2021) simultaneously minimize a task-specific loss on the source data and a language modeling

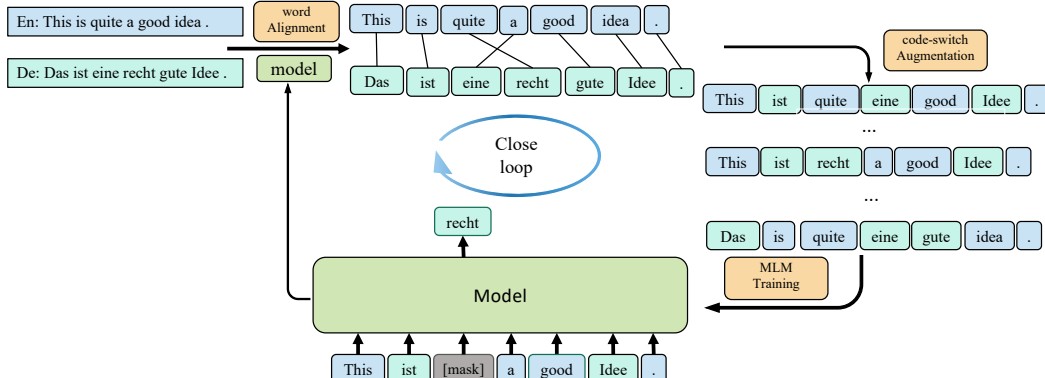

Figure 2: An overview of our iterative task-adaptive pretraining for word alignment. The alignment method is based on multilingual pre-trained LMs and self-labeled pairs obtained by alignment method will be used to augment sentences by code-switching strategy. Then multilingual pre-trained LMs will be finetuned on the augmented sentences by self-supervised learning (masked language modeling). And this process will be executed iteratively.

loss on the target data during fine-tuning. Nishida et al. (2021) propose a novel fine-tuning process: task-adaptive pre-training with word embedding regularization which runs additional pre-training by making the static word embeddings of a pretrained Language Models close to the word embeddings obtained in the target domain with fastText. Kang et al. (2022) modulate the intermediate hidden representations of PLMs with domain knowledge, consisting of entities and their relational facts. Different from them, we proposed a new paradigm, in which the output of word alignment task can be used to promote task-adaptive pretraining.

## 2.2 WORD ALIGNMENT

Statistical alignment models directly build on the lexical translation models such as the IBM models (Brown et al., 1993) and their implementations Giza++ (Och & Ney, 2003), fast-align (Dyer et al., 2013) and eflomal (Östling & Tiedemann, 2016) are widely used for alignment. Based on NMT models (Bahdanau et al., 2015) trained on parallel corpora, researchers have proposed several methods to extract alignments from them (Cohn et al., 2016; Zenkel et al., 2019; Garg et al., 2019; Chen et al., 2021; Zenkel et al., 2020a; Chatterjee et al., 2022). Cohn et al. (2016) and Zenkel et al. (2019) create alignments from attention matrices. Garg et al. (2019) extract discrete alignments from the attention probabilities learnt during regular neural machine translation model training and leverage them in a multi-task framework to optimize towards translation and alignment objectives. Zenkel et al. (2019; 2020a) both extend the network with an additional alignment layer. Chen et al. (2021) propose a self supervised word alignment model that takes advantage of the full context on the target side. Chatterjee et al. (2022) propose a simple architectural modification to modern NMT systems to obtain accurate online alignments. However, most NMT-based methods require sufficient amount parallel data to train high quality NMT systems. which limits their application in the low-resource languages and in domain-specific scenarios without extra parallel data. Sabet et al. (2020) propose effective methods to extract alignments from multilingual contextualized embeddings (Devlin et al., 2019b; Conneau et al., 2020) for word alignments without explicit training on parallel data. Dou & Neubig (2021) indicate that finetuning pre-trained LMs on extra parallel corpus can improve alignment quality. There are also work on supervised neural word alignment (Stengel-Eskin et al., 2019; Nagata et al., 2020). For example, (Nagata et al., 2020) present a novel supervised word alignment method based on cross-language span prediction and formalize a word alignment problem as question answering task. However, supervised data are not always accessible, making their methods inapplicable in many scenarios.

## 2.3 CODE-SWITCHING

Code-switching is a prevalent phenomenon in multilingual communities where the words, morphemes and phrases from two or more languages are switched in speech or writing. And it has been employed to improve NMT tasks (Yang et al., 2020a; Liu et al., 2021; Yang et al., 2020b) and cross lingual tasks (Qin et al., 2020; Zhang et al., 2021; Lee et al., 2021). Most of related work attribute the improvement of model performance to an intuitive assumption: Using code-switching data to train models will encourages them to align representations from source and target languages by mixing their context information. In this work, we dig deeper and give an approximate formal explanations for the connection between word alignment and masked language modeling.

## 3 METHOD

Although multilingual contextualized embeddings of pre-trained LMs can be employed to achieve reasonable performance even in the absence of explicit training on parallel data, there is still a clear gap: the model is trained with language modeling loss function and tested on word alignments task. Dou & Neubig (2021) leverage pre-trained LMs and fine-tune them on parallel texts with new objectives designed to improve alignment quality. However, they need extra large amounts of parallel sentences, which is at least thousands of times larger than test sentences and limit their applications on low-resource languages and settings without parallel text. So here is a practical and valuable setting: if we only have a few parallel sentences which need to be aligned and a pre-trained LM, can we further improve the performance of alignment methods? Figure 2 illustrates an overview of our paradigm. More accurate alignment results in higher-quality code-switched sentences. And finetuning on higher-quality code-switched sentences will encourage pretrained LMs to align representations from source and target languages by mixing their context information. A better pretrained LMs will obviously improve the accuracy of alignment. This iterative process will promote each other. In the following paragraphs, we will elaborate on each part.

### 3.1 WORD ALIGNMENT

The task of word alignment can be defined as: Given a source-language sentence $\mathbf{x} = \langle x_1, \cdots, x_n \rangle$ of length $n$ and its target-language translation $\mathbf{y} = \langle y_1, \cdots, y_m \rangle$ of length $m$, the method of word alignment needs to find a set of pairs of source and target words:

$$A = \{\langle x_i, y_j \rangle : x_i \in \mathbf{x}, y_j \in \mathbf{y}\} \tag{1}$$

Aligned words are assumed to correspond to each other, i.e. for each word pair $\langle x_i, y_j \rangle$, $x_i$ and $y_j$ are semantically similar to each other within the context of the sentence. We focus on improving the methods which can leverage multilingual pre-trained LMs (Devlin et al., 2019b; Conneau et al., 2020) for word alignments by extracting alignments from similarity matrices induced from their contextualized embeddings without relying on parallel data. For each pair of parallel sentences $\mathbf{x}$ and $\mathbf{y}$, these methods extract the hidden states of the $k$-th layer of the multilingual model: $h_{\mathbf{x}}^k = \langle h_{x_1}^k, \cdots, h_{x_n}^k \rangle$ and $h_{\mathbf{y}}^k = \langle h_{y_1}^k, \cdots, h_{y_m}^k \rangle$. Given these contextualized word embeddings, there are many methods to obtain alignments. For example, a simple and effective method Argmax (Sabet et al., 2020) is to align $x_i$ and $x_j$ when $h_{x_i}^k$ is the most similar to $h_{y_j}^k$ and vice-versa. That is, we set $A_{ij} = 1$ if $\left(i = \arg\max_l S_{l,j}^k\right) \wedge \left(j = \arg\max_l S_{i,l}^k\right)$ and $A_{ij} = 0$ otherwise. And $S_{ij}^k = sim\left(h_{x_i}^k, h_{y_j}^k\right)$ is some normalized measure of similarity, e.g., cosine-similarity. Some other methods frame alignment as an assignment problem (Sabet et al., 2020) or regularized optimal transport problem (Dou & Neubig, 2021; Chi et al., 2021) and is defined by $A = argmax_{A \in \{0,1\}^{l_e \times l_f}} \sum_{i=1}^{l_e} \sum_{j=1}^{l_f} A_{ij} S_{ij}$.

In our setting, these methods will self-label parallel sentences. In order to get high quality aligned pairs, we filter the pairs with a particular threshold $\epsilon$:

$$A_{filter} = \left\{\langle x_i, y_j \rangle : S_{ij}^k > \epsilon\right\} \tag{2}$$

---

**Algorithm 1:** Augmented Sentences Sampling

---

**input** : A pair of parallel sentences $\langle \mathbf{x}, \mathbf{y} \rangle$, alignment of the $t$-th iteration $A_t$, alignment of the $(t + 1)$-th iteration $A_{t+1}$, sampling probability $P_{old}$ for old pairs, sampling probability $P_{new}$ for new pairs, the sampling rounds $rounds$.

**output** : Augmented sentences $S_{aug}$

---

1   $\mathbf{S}_{aug} \leftarrow \emptyset$
2   $\mathbf{A}_{old} \leftarrow A_t \cap A_{t+1}$
3   $\mathbf{A}_{new} \leftarrow A_{t+1} \setminus (A_t \cap A_{t+1})$
4   $SET_{AP} \leftarrow \{(\mathbf{A}_{old}, P_{old}), (\mathbf{A}_{new}, P_{new})\}$
5   **for** $i \leftarrow 1$ **to** $rounds$ **do**
6     **for** $(\mathbf{A}, P) \in SET_{AP}$ **do**
7       **for** $pair \in \mathbf{A}$ **do**
8         **if** $P \geq random.random()$ **then**
9           $\langle \mathbf{x}, \mathbf{y} \rangle$ =Code-switch( $\langle \mathbf{x}, \mathbf{y} \rangle$,$pair$)
10           $\mathbf{S}_{aug} \leftarrow \mathbf{S}_{aug} \cup \langle \mathbf{x}, \mathbf{y} \rangle$

---

## 3.2 CODE-SWITCHED AUGMENTATION

Code-switching is a prevalent phenomenon in multilingual communities where the words, morphemes and phrases from two or more languages are switched in speech or writing. The switched ones usually are semantically similar.

Suppose we get $p$ different pairs of source and target words for sentence $\mathbf{x} = \langle x_1, \cdots, x_n \rangle$ and $\mathbf{y} = \langle y_1, \cdots, y_m \rangle$, it is easy to augment sentences by code-switching. The source words and target words in one pair can be considered as synonyms and can be switched. If we augmented sentences by code-switch sentences with one token at a time, then we have $C_p^1$ kinds choices, which means we can get $C_p^1$ different sentence pairs at most. If the number of switched pairs range from 0 to $p$, then the maximum of different sentence pairs without considering special cases:

$$C_p^0 + C_p^1 + C_p^2 + \cdots + C_p^p = 2^p \tag{3}$$

Obviously this is an exponential and impressive data augmentation method. In practice, we do not exhaust all code-switched sentences and the sampling method is illustrated in Algorithm 1. We set $A_0 = \emptyset$ and $A_1 = \emptyset$ so that when $t = 0$, Algorithm 1 still works and in this setting, the 0-th iteration is exactly the standard task-adaptive pretraining. When $t > 1$, alignment results of two successive iterations will be used so that we can assign different sampling distributions to the intersection and new aligned pairs. In general, we give a higher probability for new aligned pairs. In addition, we will also add origin parallel sentences to augmented dataset which is prepared for masked language modeling.

## 3.3 MLM ON CODE-SWITCHED DATASET

For masked language modeling (MLM), the input sequence $\mathbf{x}$ consists of multiple individual tokens. A fraction of the input tokens are chosen randomly and replaced with $< MASK >$ tokens. Assume that these masked indices are collected together in a set $mask(\mathbf{x})$ and we use $\hat{x}$ to denote a masked token. Then model with parameters $\theta$ learns to predict $mask(\mathbf{x})$ by the surrounding unmasked tokens $\mathbf{x}_{\backslash mask(\mathbf{x})}$.

$$\mathcal{L}_{\text{MLM}} = - \sum_{\hat{x} \in mask(\mathbf{x})} \log \mathbb{P}\left(\hat{x} \mid \mathbf{x}_{\backslash mask(\mathbf{x})}; \theta\right) \tag{4}$$

Using code-switched and origin parallel sentences to train model will encourages them to align representations from source and target languages by mixing their context information. More importantly, self-labeled word pairs have the same surrounding tokens. So training model on these sentences will obviously align code-switched tokens in implicit manner and the predicted pairs will also move towards each other in the embedding space. For example, in Figure 2, "Das" and "This" is aligned

pair. When we train model on the code-switched sentence "Das is quite a good idea. " and origin sentence "This is quite a good idea. ", the word "Das" and English word "This" will move towards each other in the embedding space because they have same surrounding tokens. We try to dig deeper and give an approximate formal explanations for the connection between word alignment and masked language modeling, which can also serve as a potential explanation why code-switching can be used to improve machine translation and cross-lingual tasks.

**Proposition**: Training model with standard masked language modeling on source-language, target-language and source-target code-switched sentences is approximately optimizing:

$$\boldsymbol{c}_{x_i}^{src} \sim \boldsymbol{e}_{x_i} \sim \boldsymbol{e}_{y_i} \sim \boldsymbol{c}_{y_i}^{tgt}$$

where $\boldsymbol{c}_{x_i}^{src}$ represents the contextualized embedding of token $x_i$ in source sentence and $\boldsymbol{c}_{y_i}^{src}$ represents the contextualized embedding of token $y_i$ in target sentence. And $\boldsymbol{e}_{x_i}$ and $\boldsymbol{e}_{y_i}$ are word embeddings of token $x_i$ and $y_i$ in vocabulary. $\boldsymbol{c}_{x_i}^{src} \sim \boldsymbol{e}_{x_i}$ represents that $\boldsymbol{c}_{x_i}^{src}$ is similar to $\boldsymbol{e}_{x_i}$. See appendix for details.

We assume that this approximate similarity is sufficient to induce word alignment. Then we have:
**Corollary**: The performance based on the last layer of optimized model (with iterative task-adaptive pretraining) is better than the best results (usually the eighth layer) of baseline model (without iterative task-adaptive pretraining).

In section 4, we will test whether this corollary holds in experiments.

## 4 EXPERIMENTS

Our experiments focus on three questions: (1) To what extent can our paradigm improve word alignment across methods, models, languages and layers. (2) The effect of the number of augmented sentences and sampling probability. (3) The detailed ablation study.

### 4.1 DATASET

Our test data are a diverse set of 6 language pairs: Persian,Czech, German, French, Hindi and Romanian, always paired with English. All of them are public dataset: En-Fa(Tavakoli & Faili), En-Cs (Marecek), En-De [1] , En-Fr(Och & Ney, 2000), En-Hi and En-Ro [2]. See Table 1 for detailed statistics of datasets.

| Langs | En-Hi | En-Fa | En-Cs | En-De | En-Fr | En-Ro |
|---|---|---|---|---|---|---|
| Size | 90 | 400 | 2500 | 508 | 447 | 203 |
| $|S|$ | 1409 | 11606 | 44292 | 9612 | 4038 | 5033 |
| $|P\backslash S|$ | 0 | 0 | 23132 | 921 | 13400 | 0 |

Table 1: Statistics of Datasets. Test sentences of the six gold word alignment datasets used in our experiments: English-Hindi (En-Hi), English-Persian (En-Fa), English-Czech (En-Cs), English-German (En-De), English-French (En-Fr), English-Romanian (En-Ro). "Size" refers to the number of sentences. S is sure alignments and P is possible alignments ($S \subset P$).

### 4.2 EVALUATION MEASURES

We use Alignment Error Rate (Och & Ney, 2003) as the standard evaluation:

$$\text{AER} = 1 - \frac{|A \cap S| + |A \cap P|}{|\Lambda| + |S|} \tag{5}$$

where A is a set of predicted alignment edges, S(sure) is (sure) unambiguous alignments and P(possible) is ambiguous alignments ($S \subset P$). We report the percentage.

---

[1]http://www-i6.informatik.rwth-aachen.de/goldAlignment/

[2]http://web.eecs.umich.edu/ mihalcea/wpt05/

| Model | Method | En-Hi | | En-Fa | | En-Cs | | En-De | | En-Fr | | En-Ro | |
|-------|--------|-------|-----|-------|-----|-------|-----|-------|-----|-------|-----|-------|-----|
| | | $F_1$ | $AER$ | $F_1$ | $AER$ | $F_1$ | $AER$ | $F_1$ | $AER$ | $F_1$ | $AER$ | $F_1$ | $AER$ |
| IBM2 | fast-align | 38 | 62 | 58 | 42 | 78 | 23 | 71 | 30 | 85 | 16 | 68 | 32 |
| IBM4 | Giza++ | 48 | 52 | 57 | 43 | 82 | 18 | 78 | 22 | 92 | 9 | 69 | 32 |
| - | eflomal | 52 | 48 | 63 | 37 | 84 | 17 | 76 | 24 | 91 | 9 | 72 | 28 |
| mBERT | Argmax | 55 | 45 | 67 | 33 | 86 | 14 | 81 | 19 | 94 | 6 | 65 | 35 |
| mBERT | Iter 0 | 56.2 | 43.8 | 68.5 | 31.5 | 86.1 | 13.7 | 80.7 | 19.2 | 94.6 | 5.3 | 66.4 | 33.5 |
| mBERT | Iter 1 | 57.5 | 42.5 | 69.6 | 30.4 | 86.2 | 13.5 | 81.2 | 18.7 | 94.7 | 5.0 | 68.9 | 31.0 |
| mBERT | Iter 2 | 57.8 | **42.1** | 70.4 | **29.6** | 86.6 | **13.2** | 81.4 | **18.5** | 95.1 | **4.6** | 69.9 | **30.1** |
| XLM-R | Argmax | 61 | 39 | 71 | 29 | 87 | 13 | 81 | 19 | 93 | 7 | 71 | 29 |
| XLM-R | Iter 0 | 62.9 | 37.1 | 72.9 | 27.1 | 88.2 | 11.6 | 81.3 | 18.6 | 94.4 | 5.3 | 72.4 | 27.6 |
| XLM-R | Iter 1 | 65.5 | 34.5 | 73.9 | 26.1 | 88.9 | 10.9 | 82.6 | **17.3** | 95.0 | **4.8** | 73.9 | 26.1 |
| XLM-R | Iter 2 | 66.2 | **33.8** | 74.6 | **25.4** | 89.3 | **10.6** | 82.5 | 17.4 | 95.0 | **4.8** | 74.2 | **25.8** |
| XLM-R | Match | 59.9 | 40.1 | 68.5 | 31.5 | 80.8 | 19.8 | 76.4 | 23.6 | 87.7 | 13.6 | 69.9 | 30.1 |
| XLM-R | Iter 0 | 61.5 | 38.5 | 70.0 | 30.0 | 81.9 | 18.7 | 77.0 | 23.0 | 88.9 | 12.3 | 70.9 | 29.1 |
| XLM-R | Iter 1 | 62.9 | 37.1 | 70.8 | **29.2** | 82.5 | 18.1 | 77.4 | 22.6 | 89.5 | 11.8 | 71.6 | 28.3 |
| XLM-R | Iter 2 | 64.3 | **35.7** | 70.7 | 29.3 | 82.6 | **18.0** | 77.7 | **22.3** | 89.6 | **11.6** | 71.9 | **28.1** |
| XLM-R | Itermax | 62 | 39 | 72 | 28 | 86 | 14 | 80 | 20 | 92 | 9 | 72 | 28 |
| XLM-R | Iter 0 | 63.8 | 36.2 | 73.6 | 26.4 | 86.5 | 13.7 | 80.2 | 19.7 | 92.2 | 8.4 | 73.0 | 26.9 |
| XLM-R | Iter 1 | 66.1 | 33.9 | 75.1 | **24.9** | 87.5 | 12.6 | 81.8 | 18.4 | 93.1 | **7.3** | 74.5 | 25.5 |
| XLM-R | Iter 2 | 66.6 | **33.4** | 75.1 | **24.9** | 87.8 | **12.4** | 81.5 | **18.2** | 93.1 | 7.4 | 74.9 | **25.1** |

Table 2: Evaluation results on six datasets. Argmax, Itermax and Match are three different alignment methods (Sabet et al., 2020) based on multilingual contextualized embeddings. Results are average over different runs. Best results are in bold. (For AER, lower is better.)

## 4.3 BASE METHODS AND MODELS

Our experiments focus on three alignment method based on multilingual pretrained models: Argmax, Itermax and Match, proposed by SimAlign (Sabet et al., 2020) and we follow the default setting of repository [3] without any modification. We use the contextualized embeddings from 8-th layer. We employ two multilingual pretrained models: the multilingual BERT model (mBERT), which is pretrained on the 104 largest Wikipedia languages and XLM-RoBERTa base (Conneau et al., 2020), which is pretrained on 100 languages on cleaned CommonCrawl data (Wenzek et al., 2020). The pretrained LMs often use subword segmentation techniques (Kudo & Richardson, 2018; Sennrich et al., 2016) and the above alignment extraction methods can only produce alignments on the subword level, we follow previous work (Sabet et al., 2020; Zenkel et al., 2020b; Dou & Neubig, 2021) and consider two words to be aligned if any of their subwords is aligned. For masked language modeling, we following Devlin et al. (2019a) and use a special [MASK] token $80\%$ of the time, a random token $10\%$ of the time and the original token $10\%$ of the time to perform masking. The batch size is set to 32. Max epoch is set to 10. We use the Adam optimizer with a learning rate of $2e - 5$. Weight decay is $0.03$. We set the filtering threshold to $0.9$. When the number of iterations is 1, it only one alignment set sampling probability to $0.7$. When the number of iterations is more than 1, we set the sampling probability $P_{old}$ to $0.7$ and $P_{new}$ to $1.0$. The sampling rounds is defaults to 5.

| Probs | En-De | En-Fa | En-Hi | En-Ro |
|-------|-------|-------|-------|-------|
| 0.5 | 17.6 | 25.9 | 34.8 | 26.1 |
| 0.7 | **17.3** | **25.8** | **34.5** | **26.1** |
| 0.9 | 17.7 | 26.2 | 35.2 | 26.5 |

Table 3: AER with different sampling probabilities.

| Rounds | En-De | En-Fa | En-Hi | En-Ro |
|--------|-------|-------|-------|-------|
| 2 | 17.4 | 26.2 | 35.1 | 26.5 |
| 5 | **17.3** | 25.8 | 34.5 | **25.8** |
| 8 | **17.3** | 25.7 | 34.3 | **25.8** |

Table 4: The effect of the number of augmented sentences.

## 4.4 RESULTS

As shown in Table 2, we report the F1 and AER scores on the six language pairs. And Table 2 also includes some widely used algorithms which is not based on pretrain mdoels, such as Giza++ (Och & Ney, 2003), fast-align (Dyer et al., 2013) and eflomal (Östling & Tiedemann, 2016). It can

---

[3]https://github.com/cisnlp/simalign

| Layer | | 0 | 1 | 2 | 3 | 4 | 5 | 6 | 7 | 8 | 9 | 10 | 11 |
|---|---|---|---|---|---|---|---|---|---|---|---|---|---|
| En-Hi | Baseline | 53.4 | 58.5 | 61.9 | 57.4 | 49.7 | 45.9 | 40.4 | 40.2 | **39.4** | 42.2 | 44.4 | 46.7 |
| | Ours | 53.5 | 55.1 | 59.0 | 52.9 | 45.8 | 41.2 | 36.0 | 35.7 | 34.5 | 35.9 | 37.0 | **39.1** |
| En-Fa | Baseline | 47.7 | 50.4 | 54.2 | 47.2 | 39.2 | 34.0 | 29.9 | 29.2 | **28.9** | 30.7 | 33.6 | 37.0 |
| | Ours | 47.6 | 48.2 | 52.0 | 45.8 | 37.1 | 30.7 | 27.2 | 26.6 | 26.1 | 25.8 | 26.1 | **28.0** |

Table 5: Evaluation results of different layers.

be observed that our method improve the performance of three alignment methods on generating high-quality alignment pairs. And from section 3.2, the standard task-adaptive pretraining is equivalent to Iter 0. With the increase of iteration rounds, our method can constantly improve model performance. This prove that iterative task-adaptive pretraining is effective. In addition, there is an impressive trends: The worse the initial performance, the more the improvement. For En-Cs, En-De and En-Fr, baseline methods perform well and the AER scores are generally lower than 20 percent. After three iterations, the AER scores are reduced by about $1.5$ points on average. For En-Ro, En-Fa and En-Hi, which are relatively low-resource or different morphological language-pairs, baseline methods perform poorly and AER scores are generally greater than 30 percent.

After three iterations, the AER scores are reduced by about $4 \sim 5$ points on average. This is an exciting result, which doesn't need any additional parallel sentences and gold labels. In the following experiments, we employ the alignment method Argmax for further analysis and explore the word alignment across different layers. Figure 3 and table 5 show that our paradigm consistently improves baseline performance cross all layers. For both En-Hi and EN-Fa, the performance of last layer based on our methods consistently outperforms the best result of baseline models. So the Corollary in section 3.3 is valid and our assumption "this approximate similarity is sufficient to induce word alignment" is supported by experiments.

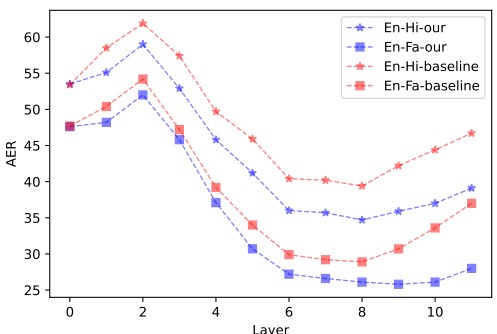

Figure 3: The comparison between ours and baseline cross layers. Lower is better.

## 4.5 ANALYSIS

We further explore the effect of sampling probability and the number of augmented sentences on model performance. We choose XLM-RoBERTa base for multilingual pretrained models and Argmax as the alignment method. Given the large amount of possible experiments when considering 6 language pairs, we do not present all scores for all languages and we will pick up four of them in most cases: En-De, an established and well-known dataset, En-Fa and En-Hi, two low-resource languages written in a different script and En-Ro. And only two rounds of iterative training are performed (t=0,1) and we only list the final results.

Table 3 shows the Alignment Error Rate in the setting of different sampling probabilities. When the sampling probability of code-switching is too high or too low, the diversity of augmented sentences will decline, which may hurt model performance. Table 3 proves this point. Although the final scores of different sampling probabilities are close, the middle probability $0.7$ achieve the best scores cross four languages. Table 4 shows the effect of the number of augmented sentences. The sampling rounds is proportional to the number of augmented sentences. From this table, we can infer that the scores will increase in general when more augmented sentences becomes available. But gains continue to decay. At the same time, the cost of MLM training will increases with more augmented sentences. So we set the sampling rounds to 5 without special statements.

| Languages | En-De | En-Fa | En-Hi | En-Ro |
|-----------|-------|-------|-------|-------|
| Baseline | 19 | 29 | 39 | 29 |
| NO-CS | 18.6 | 27.1 | 37.1 | 27.5 |
| Random | 18.4 | 26.8 | 36.5 | 27.3 |
| NO-Filter | 17.8 | 26.4 | 34.9 | 26.0 |
| Ours | **17.4** | **25.8** | **33.8** | **25.4** |

Table 6: Ablation study of code-switching strategy.

## 4.6 ABLATION STUDIES

For ablation study, we choose XLM-RoBERTa base for multilingual pretrained models and Argmax as the alignment method. We consider three kinds of ablation studies. Table 6 lists the final iterative results. "NO-CS" means only origin monolingual parallel sentences and there are no code-switched sentences. "Random" means the pairs used for code-switching are randomly generated and they are not gold-pairs in most cases. Note that the dataset "Random" also includes origin monolingual parallel sentences. "NO-Filter" means that we don't use a threshold to filter the pairs and all aligned pairs will be employed to augment code-switched sentences. The result of "NO-CS" indicates that without code-switched sentences, method can improve performance. In fact, it is almost equivalent to standard task-adaptive pretraining and Iter 0 (section 3.2). The comparison of "Random" and "NO-CS" shows that the improvement of "Random" mainly comes from origin monolingual parallel sentences. And the randomly code-switched sentences only bring a very slight boost. The comparison of "NO-Filter" and "Ours" indicates the filtering the pairs with a threshold is beneficial to model performance.

## 5 CONCLUSION

Inspired by the fact that continued pretraining of pretrained models on the unlabeled data of a given task has been show to be beneficial for task performance, we further design an iterative task-adaptive pretraining paradigm for word alignment, in which task-adaptive pretraining will be performed not only before task but also after task. The multilingual models will be continuously finetuned on the augmented code-switched dataset. The iterative process will promote each other. More accurate alignment results in higher-quality code-switched sentences. And finetuning on higher-quality code-switched sentences will encourage pretrained LMs to align representations from source and target languages by mixing their context information. A better pretrained LMs will obviously improve the accuracy of alignment. Experimental results on six language pairs and demonstrate that our paradigm can consistently improve baseline methods.

We are considering a more general paradigm about iterative task-adaptive pretraining and will apply the paradigm to other token-level tasks such as such as Named Entity Recognition and Parts-of-speech tagging. And how to establish a connection between the downstream tasks and self-supervised tasks of pretraining stage is key point, especially for low-resources tasks and languages. In fact, the prompt-based methods in which downstream tasks are reformulated to language modeling are alternative solutions. And we are trying to combine our iterative task-adaptive pretraining with prompt-based methods.

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

# A APPENDIX

## A.1 A PROOF OF MLM ON CODE-SWITCHED DATASET

**Proposition** Training model with standard masked language modeling on source-language, target-language and source-target code-switched sentences is approximately optimizing:

$$\boldsymbol{c}_{x_i}^{src} \sim \boldsymbol{e}_{x_i} \sim \boldsymbol{e}_{y_i} \sim \boldsymbol{c}_{y_i}^{tgt}$$

where $\boldsymbol{c}_{x_i}^{src}$ represents the contextualized embedding of token $x_i$ in source sentence and $\boldsymbol{c}_{y_i}^{src}$ represents the contextualized embedding of token $y_i$ in target sentence. And $\boldsymbol{e}_{x_i}$ and $\boldsymbol{e}_{y_i}$ are word embeddings of token $x_i$ and $y_i$ in vocabulary.

**Note** Under the existing conditions, we can not derive a strict bound but an approximate conclusion. Given $\boldsymbol{a} \in \mathbb{R}^n, \boldsymbol{b} \in \mathbb{R}^n, \boldsymbol{c} \in \mathbb{R}^n$ we assume $\boldsymbol{a} \sim \boldsymbol{b}$, if the projection components of $\boldsymbol{a}$ and $\boldsymbol{b}$ onto another vector $\boldsymbol{c}$ are the same: $\boldsymbol{a} \cdot \boldsymbol{c} = \boldsymbol{b} \cdot \boldsymbol{c}$. We think this is approximately reasonable for word alignment task, because for word alignments method two words are aligned as long as their similarity is higher than other words in two parallel sentences and doesn't need to exceed a fixed number. And in section 3.3, we give a corollary. Subsequent experiments prove that our assumption is reasonable.

**Proof** We denote the embeddings of the corresponding original tokens as $\boldsymbol{e}_1, \boldsymbol{e}_2, \cdots, \boldsymbol{e}_L$. The MLM objective $\mathcal{L}_{\text{MLM}}(\boldsymbol{x})$ can be formulated as:

$$-\frac{1}{|\mathcal{M}|} \sum_{i \in \mathcal{M}} \log \frac{\exp\left(\boldsymbol{m}_i \cdot \boldsymbol{e}_i\right)}{\sum_{k=1}^{|\mathcal{V}|} \exp\left(\boldsymbol{m}_i \cdot \boldsymbol{e}_k\right)} = -\frac{1}{|\mathcal{M}|} \sum_{i \in \mathcal{M}} \log \sum_{k=1}^{|\mathcal{V}|} \exp\left(\boldsymbol{m}_i \cdot \boldsymbol{e}_k - \boldsymbol{m}_i \cdot \boldsymbol{e}_i\right) \quad (6)$$

where $\mathcal{M}$ denotes the set of masked tokens and $|\mathcal{V}|$ is the size of vocabulary $\mathcal{V}$. $\boldsymbol{m}_i$ is hidden state of the last layer at the masked position, and can be regarded as a fusion of contextualized representations of surrounding tokens. Given two sentences: one source-language sentence $\mathbf{x} = \langle x_1, \cdots, x_{i-1}, x_i, x_{i+1}, \cdots, x_n \rangle$ of length $n$ and its code-switched sentence $\mathbf{x}' = \langle x_1, \cdots, x_{i-1}, y_i, x_{i+1}, \cdots, x_n \rangle$, where $\langle x_i, y_i \rangle$ is aligned pair. If we only mask $x_i$ in the $\mathbf{x}$ and $y_i$ in the $\mathbf{x}'$, then $\mathbf{x_{mask}} = \langle x_1, \cdots, x_{i-1}, <mask>, x_{i+1}, \cdots, x_n \rangle = \langle x_1, \cdots, x_{i-1}, <mask>, x_{i+1}, \cdots, x_n \rangle = \mathbf{x'_{mask}}$, the loss function can be written as

$$\mathcal{L}_{\text{MLM}} = \mathcal{L}_{\mathbf{x}} + \mathcal{L}_{\mathbf{x}'} = -\frac{1}{2} \left( \log \sum_{k=1}^{|\mathcal{V}|} \exp\left(\boldsymbol{m} \cdot \boldsymbol{e}_k - \boldsymbol{m} \cdot \boldsymbol{e}_{x_i}\right) + \log \sum_{k=1}^{|\mathcal{V}|} \exp\left(\boldsymbol{m} \cdot \boldsymbol{e}_k - \boldsymbol{m} \cdot \boldsymbol{e}_{y_i}\right) \right) \quad (7)$$

This inequality below is easily proved.

$$\max\left\{x_1, \ldots, x_n\right\} \leq \log \sum_{i=0}^{n} e^{x_i} \leq \max\left\{x_1, \ldots, x_n\right\} + \log n \quad (8)$$

So for

$$\mathcal{L}_{\mathbf{x}} = -\log \sum_{k=1}^{|\mathcal{V}|} \exp\left(\boldsymbol{m} \cdot \boldsymbol{e}_k - \boldsymbol{m} \cdot \boldsymbol{e}_{x_i}\right) \quad (9)$$

We have:

$$\max \begin{pmatrix} \boldsymbol{m} \cdot \boldsymbol{e}_0 - \boldsymbol{m} \cdot \boldsymbol{e}_{x_i} \\ \vdots \\ \boldsymbol{m} \cdot \boldsymbol{e}_{x_i-1} - \boldsymbol{m} \cdot \boldsymbol{e}_{x_i} \\ 0 \\ \boldsymbol{m} \cdot \boldsymbol{e}_{x_i+1} - \boldsymbol{m} \cdot \boldsymbol{e}_{x_i} \\ \vdots \\ \boldsymbol{m} \cdot \boldsymbol{e}_{|\mathcal{V}|} - \boldsymbol{m} \cdot \boldsymbol{e}_{x_i} \end{pmatrix} \leq \log \sum_{k=1}^{|\mathcal{V}|} e^{\boldsymbol{m} \cdot \boldsymbol{e}_k - \boldsymbol{m} \cdot \boldsymbol{e}_{x_i}} \leq \max \begin{pmatrix} \boldsymbol{m} \cdot \boldsymbol{e}_0 - \boldsymbol{m} \cdot \boldsymbol{e}_{x_i} \\ \vdots \\ \boldsymbol{m} \cdot \boldsymbol{e}_{x_i-1} - \boldsymbol{m} \cdot \boldsymbol{e}_{x_i} \\ 0 \\ \boldsymbol{m} \cdot \boldsymbol{e}_{x_i+1} - \boldsymbol{m} \cdot \boldsymbol{e}_{x_i} \\ \vdots \\ \boldsymbol{m} \cdot \boldsymbol{e}_{|\mathcal{V}|} - \boldsymbol{m} \cdot \boldsymbol{e}_{x_i} \end{pmatrix} + \log n \quad (10)$$

In Ineq.10, 0 is fixed value. So when training model with this loss function, model is optimized to learn $\boldsymbol{m} \cdot \boldsymbol{e}_k - \boldsymbol{m} \cdot \boldsymbol{e}_{x_i} \leq 0, \forall k \in |\mathcal{V}|$. In other words, $\boldsymbol{m} \cdot \boldsymbol{e}_k \leq \boldsymbol{m} \cdot \boldsymbol{e}_{x_i}, \forall k \in |\mathcal{V}|$. When $k = y_i$,

we have $\boldsymbol{m} \cdot \boldsymbol{e}_{y_i} \leq \boldsymbol{m} \cdot \boldsymbol{e}_{x_i}$. Similarly, for $\mathcal{L}_{\mathbf{x}'}$, we have $\boldsymbol{m} \cdot \boldsymbol{e}_{x_i} \leq \boldsymbol{m} \cdot \boldsymbol{e}_{y_i}$. So when training model with loss function $\mathcal{L}_{\mathrm{MLM}} = \mathcal{L}_{\mathbf{x}} + \mathcal{L}_{\mathbf{x}'}$, model will be optimized to learn $\boldsymbol{m} \cdot \boldsymbol{e}_{x_i} = \boldsymbol{m} \cdot \boldsymbol{e}_{y_i}$. This equation can't ensure $\boldsymbol{e}_{x_i} = \boldsymbol{e}_{y_i}$ but $\boldsymbol{e}_{x_i} \sim \boldsymbol{e}_{y_i}$ to some extent. For standard masked language modeling, there is a probability that the original token will not be masked and we use $\boldsymbol{c}_{x_i}^{src}$ to represent the hidden state of the last layer, which is the contextualized embedding of token $x_i$. So we have $\boldsymbol{c}_{x_i}^{src} \cdot \boldsymbol{e}_k \leq \boldsymbol{c}_{x_i}^{src} \cdot \boldsymbol{e}_{x_i}$, $\forall k \in |\mathcal{V}|$. Obviously, $\boldsymbol{e}_{x_i} \sim \boldsymbol{c}_{x_i}^{src}$. Similarly, if we consider target language sentence $\mathbf{y} = \langle y_1, \cdots, y_{i-1}, y_i, y_{i+1}, \cdots, y_n \rangle$, we have $\boldsymbol{e}_{y_i} \sim \boldsymbol{c}_{y_i}^{tgt}$. So training model with masked language modeling on source-language, target-language and source-target code-switched sentences is approximately optimizing:

$$\boldsymbol{c}_{x_i}^{src} \sim \boldsymbol{e}_{x_i} \sim \boldsymbol{e}_{y_i} \sim \boldsymbol{c}_{y_i}^{tgt}$$

