# OpenReview forum: "Iterative Task-adaptive Pretraining for Unsupervised Word Alignment"
_ICLR.cc/2023/Conference — Submitted to ICLR 2023_

### Official Review · Reviewer_hkhx · 2022-10-23

**Confidence:** 4
**Correctness:** 3
**Technical Novelty And Significance:** 2
**Empirical Novelty And Significance:** 2
**Recommendation:** 3

**Clarity, Quality, Novelty And Reproducibility:**


See "Strength And Weaknesses".

**Strength And Weaknesses:**

Strengths:
- the paper is well-written and illustrated
- the idea is very simple
- the application to word alignment is novel
- the method is somewhat reproducible

Weaknesses:
- the paper lacks of substance. To me, it only sticks together some pieces from previous work (task-adaptative pre-training) and applies it to a particular task.
- the need for code switching is never properly motivated in the paper. It makes sense to do code switching, but since this is a crucial point of the method it should be extensively motivated. Why does the method use code-switching? Code-switching is introduced very early, but briefly, in the paper, then there is a section about it in related work, and then it is presented as a component of the method. The relationship between task-adaptative pretraining and word alignments is clearly given and motivated, but we miss the links with code-switching.
- the analysis doesn't answer the questions that I expect the readers will have:
- How does this method increase the computational cost of the training? (it looks like a huge increase)
- How does this method compare to just pre-training XLM-R for a few more epochs? (I have no idea)
- What are the limits of the method?




**Summary Of The Paper:**

This proposes an  iterative self-supervised task-adaptive pretraining  framework with a focus on the task of word alignment. In other words, task adaptative pretraining and word alignments are done iteratively. The method works without labeled data.
Experiments shows a reduction of the AER with an XLM-R model.


**Summary Of The Review:**

In my opinion the proposed method lacks of originality. In short, I would described it as an application of task-adaptative pre-training to the task of word alignments. An extensive analysis and better motivation for code switiching would improve the paper, but it would still lack of substance for a top-tier conference.

---

> ### Author Response · Authors · 2022-11-14
> **Reply to Weaknesses**
>
> Thank you for your comments. But we have to point out that you have some misunderstandings about our work and ignore some key points.
>
> ### Reply to Weakness 1 & 4:
>
> “the paper lacks of substance. To me, it only sticks together some pieces from previous work (task-adaptative pre-training) and applies it to a particular task.” “How does this method compare to just pre-training XLM-R for a few more epochs? (I have no idea)”
>
> Pre-training refers to using self-supervised loss function to train the model on large-scale corpus. In order to improve the adaptation of pretrained models on domain/task, domain/task adaptive pre-training employ self-supervised loss functions to continue training pretrained models on domain/task datasets. However, we think the standard task-adaptative pre-training doesn’t t sufficiently integrate pre-training with the word alignment task. Standard task-adaptative pre-training only occurs before tasks. There is an obvious lack of interactive feedback. Can the output of the task can be used by pretraining? So we propose an iterative self-supervised continued pretraining paradigm, constantly pushing pre-trained LMs toward the word alignment task.
> The standard task-adaptative pre-training can be regard as a special case (only iterate once) of our method.  In the last paragraph of section 3.2 ,  we emphasize that  “the 0-th iteration is exactly the standard task-adaptive pretraining.” From you question “How does this method compare to just pre-training XLM-R for a few more epochs? (I have no idea)”, we suspect that you miss or ignore this key point. See table 2. And the results across methods, models, and datasets can be found at Iter 0.
>
> ### Reply to weakness 2:
>
> We are happy that you can understand the relationship between task-adaptative pretraining and word alignments.
> Section 3.2 CODE-SWITCHED AUGMENTATION show the relationship between the code-switching and word alignment pairs. In brief, we can't do code-switching augmentation without word alignment and code-switching only happens between aligned tokens.
> Section 3.3 MLM ON CODE-SWITCHED DATASET show the relationship between the code-switching and pretraining.
> "Why does the method use code-switching?"  Without code-switching sentences, pure task-adaptive pretraining can’t continuously improve model performance. And this is the key point why we can design the Iteration mechanism. In Section 3.3, we also give an example: When we train model on the code-switched sentence "Das is quite a good idea. " and origin sentence "This is quite a good idea. ", the word "Das" and English word "This" will move towards each other in the embedding space because they have same surrounding tokens.
> At the same time, code-switched sentences play an important role in the proof of Proposition.
>
>
>
> ### Reply to weakness 3:
>
> "How does this method increase the computational cost of the training? (it looks like a huge increase)"
>
> Although our code-switched augmentation algorithm theoretically is exponential and impressive (see section 3.2), we only augment the datasets by 5 times(see sampling rounds in section 4.3 ). The original datasets usually contain hundreds of sentences,so the
> the actual computational overhead is not huge. There is no change from the standard MLM in terms of computation complexity. And we report the average computation time of one round of iteration with only one RTX A6000.
>
> |language-pairs | seconds|
> |--|--|
> |En-Hi |32.060245|
> |En-Fa |116.321279|
> |En-Cs |1027.55578|
> |En-De |145.265363|
> |En-Fr |97.4777085|
> |En-Ro |209.874575|
>
> And we think the computation time is acceptable. Just wait for several minutes and a better result is coming soon. **There‘s no such thing as a free lunch.**  Without any extra parallel sentences, if we want to improve performance, we have to pay some other things, which is additional computing resources here.
>
>
> ### Reply to weakness 5:
>
> "What are the limits of the method?"
>
> You have already pointed out its limitation. This method needs additional computational cost compared with basline (Sabet et al. (2020)).
>
>
> Maybe you are not familiar with cross-lingual/multilingual. It seems that you didn't understand the paper very well before. If our replies can address your confusion, please reconsider the recommendation score. Thanks a lot !

---

### Official Review · Reviewer_j4s3 · 2022-10-24

**Confidence:** 3
**Correctness:** 3
**Technical Novelty And Significance:** 2
**Empirical Novelty And Significance:** 1
**Recommendation:** 5

**Clarity, Quality, Novelty And Reproducibility:**

This paper is easy to follow. However, it is a little bit unclear about how the hyperparameters were tuned given that a single dataset is employed for tuning and testing without a separate development (or validation) data.

**Strength And Weaknesses:**

Strength

* It is a simple yet effective method to improve word alignment quality without any supervisions.

Weakness

* It is not clear why the proposed method works. The author tries to prove it by the similarity of code-switched and masked sentences. However, the proposed method might prone to poor local optimum problem in the similar manner as done by an EM algorithm.

* The experimental procedure is a bit unclear in that that there exists no training data, but a model is simply fine-tuned on test data. Thus, it is not clear how the hyper-parameters are tuned, e.g., sampling probability, for a fair comparison.

* Given the iterative procedure for fine-tuning, it is not clear how the word alignment will be revised, e.g., whether the proposed method is trying to concentrate on more confident pairs or not. Probably it would be good to analyze the word alignment confidence in each iterations.

**Summary Of The Paper:**

This work propose a simple fine tuning method for unsupervised word alignment based on multilingual contextualized pre-trained language model, e.g., BERT. This method is based on the similarity score of the word representations in two languages and employs fine tuning by 1) iteratively augmenting code-switched training data based on the previously predicted word alignment and 2) masking inputs. The proposed method allows a model to focus more on the paired word representation considering context. Experimental results show that the proposed method achieved gains when compared to the prior baselines.

**Summary Of The Review:**

This work is interesting in employing code-switching during fine-tuning combined with masking. Experiments show gains when compared with baselines. However, it is not clear how the hyperparamters were tuned.

---

> ### Author Response · Authors · 2022-11-14
> **Reply to weakness 1,2,3**
>
> We thank the reviewer for the valuable comments.
>
> ### Reply to weakness 1:
>
> A high filtering threshold is the key.  It seems that you miss something important. See equation (2). ” In our setting, these methods will self-label parallel sentences. In order to get high quality aligned pairs, we filter the pairs with a particular threshold ϵ.”
> See section 4.3 BASE METHODS AND MODELS, “We set the filtering threshold to 0.9.” Alignment pairs usually are correct if they have high degree of confidence. When we use high confidence prediction pairs to augment the code-switch dataset, the model will converge to the global optimal rather than the local optimal. If we do not select the alignment pairs with high confidence rate, but directly select all alignment pairs, it will indeed easily fall into the local optimal solution as you speculated. Refer to section 4.6 ABLATION STUDIES."NO-Filter" setting can be seen as a local optimal.
>
>
> ### Reply to weakness 2:
>
> On the one hand, many word alignment dataset often don’t contain the dev dataset, this is where the pretrained model-based word alignment task differs from other tasks. Previous work usually report the effect of different hyper-parameters on test dataset.
> For example, in our baseline method Sabet et al. (2020), see section 2.2, they introduce two hyper-parameters for “Distortion and Null”. They only report the results of different values on test dataset.  Dou and Neubig (2021) also conduct a analysis on the threshold c for softmax alignment extraction method and report the results of different values on test dataset and pick the best one for main experiment.
> So for word alignment task showing the effects of different hyperparameters on test dataset is acceptable by other peer reviewers.
>
> On the other hand, sampling probability relates to the diversity of the augmented samples.
> If the sampling probability is too small, then the augmented sentence will be close to the source sentence, if the sampling probability is too large, then the augmented sentence will be close to the target sentence. Ideally, all words should have themself alignment pairs. Setting sampling probability to 0.5 ensures maximum diversity. But in our method, not all alignment pairs will be used to augment sentences, we filter them will a particular threshold ϵ. So sampling probability should be slightly bigger than 0.5. In practice, see Table 3, we recommend a range of 0.5 to 0.7. For a fair comparison, our setting about sampling probability is same for all datasets.
> When the number of iterations is 1, it only one alignment set sampling probability to 0.7. When the number of iterations is more than 1, we set the sampling probability P_{old} to 0.7 and P_{new} to 1.0.
>
> For other hyperparameters, during masked language modeling, we adopt standard setting. As claimed in section 4.3 BASE METHODS AND MODELS, the batch size is set to 32. Learning rate is 2e−5. Weight decay is 0.03. Max epoch is set to 10 and the last checkpoint will be used for test.  The two new hyperparameters are sampling probability and sampling round. Sampling probability relates to the diversity of the augmented samples, and sampling round determines the number of augmented samples. For sampling round, see Table 4, considering the trader-off between computation resources and time, we recommend 5. We are planning to release code to help practitioners
>
>
> ### Reply to weakness 3:
>
> Yes. Refer to replies for Weakness 1. Following your advice, we further analyze the details in each iterations. (Model：XLM-R. Dataset：En-Hi )
> |Iter|The number of pairs|Average confidence|
> |--|--|--|
> |Iter0| 786 | 0.9132|
> |iter1| 831|0.9165|
> |iter2|839|0.9217|
> The model becomes more “confident and optimistic”, tends to predict more alignment pairs and increases the average confidence rate of all predicted alignment pairs.
>
>  If our answer is satisfactory to you, please reconsider the recommendation score. Thanks  a lot !

---

### Official Review · Reviewer_KcR5 · 2022-10-25

**Confidence:** 2
**Correctness:** 4
**Technical Novelty And Significance:** 3
**Empirical Novelty And Significance:** 3
**Recommendation:** 6

**Clarity, Quality, Novelty And Reproducibility:**

The paper is mostly clear with a few typos. The paper has solid quality and is novel. I do not see any reproducibility problem.

**Strength And Weaknesses:**

Strength:
- The proposed method is well-motivated and clear.
- Empirically, the method can effectively improve alignment error rate.

Weakness:
- The biggest weakness of the proposed method is that it can be slow. Practitioners will need to decide whether using the method is worth the computation time and how to choose hyperparameters. Therefore, it would be very helpful if the paper can discuss computation time and add relevant metrics in the experiment result tables.
- The introduction suggests that the proposed method (or framework) can be applied to other NLP tasks, but the paper is restricted to only one task (word alignment). Ideally, the paper should have other tasks to show that this framework can really generalize to other tasks.
- The paper only experiments with up to 3 iterations. I am curious if the method can continually improve AER. Does the method converge after 3 iterations?

**Summary Of The Paper:**

This paper studies unsupervised word alignment using pre-trained multilingual LMs. The paper proposes an iterative method that repeats two steps: (1) generate word alignment using a LM, (2) using the word alignment, generate synthesis code-switched parallel text to fine-tune the LM. Empirically, this iterative method reduces alignment error rate for six language pairs.

**Summary Of The Review:**

The paper proposes a novel method for unsupervised word alignment and shows good results. However, the paper does not discuss time complexity, which could be a key weakness of the method. Overall, I weakly recommend acceptance.

---

> ### Author Response · Authors · 2022-11-14
> **Reply to weakness 1,2,3**
>
> We thank the Reviewer for positive comments and helpful feedback on our work.
>
>
> ### Reply to weakness 1:
>
> **There‘s no such thing as a free lunch**.  Without any extra parallel sentences, if we want to improve performance, we have to pay some other things, which is additional computing resources here. Following your advice, we static the average computation time of one round of iteration with only one RTX A6000.
>
> |language-pairs | seconds|
> |--|--|
> |En-Hi |32.060245|
> |En-Fa |116.321279|
> |En-Cs |1027.55578|
> |En-De |145.265363|
> |En-Fr |97.4777085|
> |En-Ro |209.874575|
> And we think the computation time is acceptable. Just wait for several minutes and a better result is coming soon. For hyperparameters, during masked language modeling, we adopt standard setting. As claimed in section 4.3 BASE METHODS AND MODELS, the batch size is set to 32. Learning rate is 2e-5. Weight decay is 0.03. Max epoch is set to 10 and the last checkpoint will be used for test.  The two new hyperparameters are sampling probability and sampling round. Sampling probability relates to the diversity of the augmented samples, and sampling round determines the number of augmented samples.
> For sampling probability, see Table 3, in practice, we recommend a range of 0.5 to 0.7
> For sampling round, see Table 4, considering the trader-off between computation resources and time, we recommend 5. We are planning to release code to help practitioners.
>
> ### Reply to weakness 2:
>
> We don’t claim that “our proposed method is a general framework” in introduction or regard ti as a core contribution. You can see that all statements are related to the word alignment task in our paper, although our proposed paradigm may indeed have the potential to become a general framework.  Just in conclusion, for future work, we mention that in “We are considering a more general paradigm about iterative task-adaptive pretraining and will apply the paradigm to other token-level tasks such as such as Named Entity Recognition and Parts-of-speech tagging.” If there is any overclaim, please point it out explicitly.
>
> ### Reply to weakness 3:
>
> In theory, the method can indeed continue to improve model performance as the number of iterations increases. Because predictions with high confidence rates can help to train better models, good models produce better predictions. However, after 3 iterations, most models on different datasets will converge gradually. See table 2 in paper. The absolute gain of continuing iterative training becomes smaller and smaller. Therefore, given the balance of computing resources and model performance, it makes little sense to continue the iteration.
>
>
> Thanks again ! Any other questions are welcome !

---

### Official Review · Reviewer_mPaq · 2022-10-26

**Confidence:** 5
**Correctness:** 4
**Technical Novelty And Significance:** 2
**Empirical Novelty And Significance:** 3
**Recommendation:** 5

**Clarity, Quality, Novelty And Reproducibility:**

The high-level idea of this paper is easy to understand, but some important details are not clear.
The novelty of the proposed method is not very good.

**Strength And Weaknesses:**

Strengths:

1. The proposed approach is appealing and easy to implement.
2. The experiment results shows that the proposed approach achieves good performance compared with the baseline.


Weaknesses:


1. Although the key idea for the proposed is explained clearly, some important details are unknown such that I have some key concerns on the proposed approach. Specifically, is the equation (4) updated at the sentence level or updated on the entire dataset? By the sentence level, I mean that the model is updated only for each sentence x and then the updated model is applied for inference on this sentence only. I guess it is updated on the entire dataset and then the natural question is how this approach works on a very small test dataset, for example, including one or two sentences. In addition, how many sentences does this method need in the test dataset to achieve good performance? These questions are important to show the superiority of the proposed method, but this paper does not study them.

2. In essence, the technical contribution of the proposed approach is limited in my opinion. At the high level, this approach is similar to Dou and Neubig (2021): both of them are extensions of Sabet et al. (2020); both employ the multi-lingual pretrained language model as the initialization of the word alignment model, and require a bilingual dataset to optimize another loss function which is aware of word alignment. The difference is that this paper only requires a relative small bilingual dataset (i.e., test dataset) whereas Dou and Neubig (2021) uses a large bilingual dataset. Theoretically, it is possible to use the similar idea of Dou and Neubig (2021) on the same scenario as a baseline.


**Summary Of The Paper:**

This paper considers a challenging scenario for unsupervised word alignment where   human labeled word alignment data is not available and only a bilingual dataset for testing is given. Technically, it extends the alignment model proposed by Sabet et al. (2020) and iteratively updates the alignment model defined on top of a multi-lingual pre-trained language model. Specifically, it iteratively performs the follow two steps: 1. it employs the updated alignment model to generate some word alignment dataset with pseudo alignments for the given bilingual test dataset; 2. it fine-tunes the multi-lingual pre-trained language model through code-switch to updates the word alignment model. In this way, the multi-lingual pre-trianed model can be optimized for the word alignment task. Experiments on five datasets demonstrate the proposed approach delivers consistent improvements over the baseline Sabet et al. (2020).

**Summary Of The Review:**

Please strengths and weaknesses.

---

> ### Author Response · Authors · 2022-11-14
> **Reply to Weakness 1,2**
>
> We thank the reviewer for the valuable comments.
>
> ### Reply to weakness 1:
>
> To make it easier to compare with basline Sabet et al. (2020), the equation (4) is updated on the entire dataset in our experiment. And we think that your suggestion is quite meaningful because in practice the alignment sentences may be very few. Thanks a lot! So we further conduct experiments on English-Hindi dataset and split the whole dataset (contains 90 different sentences) into 90 different datasets. Each dataset consists of one sentence and we conduct experiments on these datasets with XLM-R, respectively. After the iterations, the average AER score decreases from 39 to 32.9, and if the equation (4) is updated on the entire dataset (refer to table 2) , the score is 33.8.  This indicate that our method indeed has better performance on instance-level or few-shot/low-resource setting. We randomly chose 10 sentences of English-Hindi dataset to present here.
>
> | Sentence-id | Baseline |Ours |
> |--|--|--|
> |26| 51.2| 44.2|
> |32| 23.1| 7.7|
> |42| 0.0|  0.0|
> |45 |76.5| 52.9|
> |58 |25.7| 22.2|
> |60 |100.0| 50.0|
> |68 |13.3|13.3|
> |72 |0.0|0.0|
> |86 |15.8|11.1|
> |87 |27.3| 27.3|
> |Average |33.3 |22.9|
> The average decrease of AER 10.39=33.3-22.9 is even greater. In total, our paradigm can significantly improve model performance with high probability. And when there is less data, the improvement is likely to be more significant.  This happens not only within one language pair, but also between different language pairs. For example, the datasets of En-Hi and En-Ro are smaller than En-Cs and En-De (refer to table 1). But the formers have more gains in table 2. For the underlying reasons, we assume that for the alignment method based on pre-training model, the model's adaptation to specific sentences is more important than the promotion between different sentences.
>
>
>
>
> ### Reply to weakness 2:
>
> Thank you for your suggestion that using Dou and Neubig (2021) to finetune the multilingual model on the test dataset as a baseline. To be honest, we also hope to compare with more baselines to prove the effective of our method. But previous methods, such as Dou and Neubig (2021), need an extra and large bilingual dataset, which obviously limit their application on low-resource language-pairs (En-Hi and En-Ro). This is different from our setting. But you inspire us and following your advice, we further supplement the experiment and finetune model on test dataset with Dou and Neubig (2021)
> The AER scores of four datasets are listed as below:
>
> |Method|En-Hi |En-De|En-Fr|En-Fa|
> |--|--|--|--|--|
> |Dou (2021) | 43.3 | 18.9| 5.0|29.5|
> |Ours| 42.1|18.5|4.6|29.6|
> From the results above, we can see that for three language-pairs (En-Hi, En-De, En-Fr), our method is better than Dou (2021). This proves that MLM with our iterative task-adaptive pretraining paradigm is better than the more elaborate loss functions (Dou (2021)).
>
> Your suggestions are really valuable! Thanks again！ If our replies can satisfy you, please reconsider the recommendation score.

---

### Author Response · Authors · 2022-11-17
**General comments to reviewers**

As the deadline for discussion period is approaching, we would like to request the reviewers to let us know whether the rebuttal sufficiently addresses the raised concerns. If there are further concerns, we would be glad to address them as quickly as possible. We are looking forward to the response from the reviewers.

---

### Decision · Program_Chairs · 2023-01-20

**Decision:**

Reject

**Justification For Why Not Higher Score:**

None of the reviewers were excited about this paper and 3/4 felt negatively. Rejection is recommended based on reviewer consensus.

**Justification For Why Not Lower Score:**

N/A

**Metareview: Summary, Strengths And Weaknesses:**

Reviewers thought the paper is well-written and easy to understand. They questioned its novelty (mPaq, hkhx) and practicality due to increased training expense (KcR5, hkhx). They suggest more baselines, especially comparisons to similar code-switching based methods. Showing benefits for an end task other than word alignment may also be useful, perhaps even a task that has code-switching in itself.

The authors responded with experiments addressing comparisons to baseline (results are fairly similar to Dou 2021), and computational experiments addressing the additional time complexity. The reviewers are unconvinced (or did not revisit the paper).